# TAKE: Task-Aware Chunked KV Cache Eviction for Efficient Long-Context LLM Prefill

## Abstract

The rapid development of large language models (LLMs) enhances various language generation applications, but it remains a serious memory usage challenge in long-context inference. Existing global pruning aims to reduce memory in the decoding process, ignoring the prefill peaks to delay the time-to-first-token. In this paper, we present **Task-Aware Chunked KV Cache Eviction** (TAKE), a training-free framework to optimize KV cache memory during the prefill stage of LLM inference. TAKE partitions long sequences into chunks and incrementally performs task-aware KV fusion and eviction, thereby avoiding full-sequence processing and reducing memory and compute overhead. To preserve task-relevant information, we introduce lightweight task-aware probe tokens to identify salient tokens within each chunk and accumulate semantic information across chunks. Furthermore, we propose a delayed eviction strategy that protects shallow transformer layers from early pruning, mitigating representation degradation and improving performance stability. Extensive experimental results show that TAKE achieves superior performance, reduces the peak GPU memory usage for the KV cache and activation to about 8.9% of the baseline model, and lowers first-token latency by over 60% for sequences up to 128k tokens. It also enables stable inference with arbitrary length contexts on 24GB consumer GPUs without quantization or KV offloading, while maintaining model quality. Our code is available at Source Code of TAKE.

## 1 Introduction

The advancements of large language models (LLMs) have significantly improved performance across diverse language tasks. However, it also poses challenges in computational efficiency, particularly for long-context inference scenarios such as document-level grounding and multi-turn dialogue modeling (Jin et al., 2025; Wang et al., 2024; Li et al., 2024a). Recent state-of-the-art models (Meta AI, 2025; Qwen Team, 2025) support context windows up to 128k tokens or more, greatly expanding the potential for long-sequence understanding, but also exacerbating the bottlenecks in memory usage and latency. A major challenge arises during the prefill stage of inference, where the model encodes all input tokens before generating any output. The transformer-based LLMs store a distinct KV cache for each token at every layer, resulting in substantial memory usage and increased risk of out-of-memory (OOM), especially on edge devices or cost-sensitive deployment environments.

To mitigate these issues, some approaches design novel architectures to compress the KV cache during model pretraining. For example, MLA (Liu et al., 2024a) applies low-rank decomposition, while MoBA (Lu et al., 2025) routes queries to dedicated KV partitions to reduce redundancy. Although these techniques achieve effective performance, they require custom model architectures and pretraining, limiting their applicability to existing LLMs. Consequently, training-free KV eviction strategies offer broader compatibility and lower deployment costs. H2O (Zhang et al., 2023) prunes tokens based on attention weights during prefill, retaining only critical KVs for decoding, and SnapKV (Li et al., 2024b) introduces an observation window to improve retention accuracy. However, these methods typically apply global, post-hoc pruning within each layer after full attention computation, failing to reduce peak memory usage during the prefill stage and remaining vulnerable to OOM under strict resource budgets.

Figure 1: Comparison of global KV eviction and chunked KV eviction strategies at prefill stage.

To address these limitations, we propose a novel **T**ask-**A**ware Chunked **K**V Cache **E**viction (TAKE) method that enables efficient prefill for long-context LLM inference while preserving LLM performance. As illustrated in Figure 1, TAKE departs from global token eviction by partitioning the entire sequence into several chunks and computing attention only within the current chunk and accumulated lightweight summaries of previous chunks. This design drastically reduces memory and computation requirements during prefill. TAKE is built on three key components: (1) Inspired by chunked prefill, the input sequence is incrementally processed in chunks, with KV caching and pruning interleaved across chunks, ensuring that the model never holds the full tokens in memory. (2) A task-aware probe token with accumulated information of previous tokens is inserted into the current chunk to identify salient tokens across chunks and fuse probe features for task-driven KV selection. (3) A delayed pruning is applied for shallow layers and immediate eviction is for deep layers and subsequent chunks, while retrospectively using mid-layer token indices to prune shallow-layer KV caches to minimize forward-pass information loss.

We evaluate TAKE on long-context benchmarks using LLaMA-3.1-8B-Instruct and Mistral-7B-Instruct-v0.3, including Needle-in-a-Haystack (NIAH) and LongBench. TAKE consistently reduces prefill memory usage and first-token latency (TTFT), while maintaining strong performance across tasks. Notably, without quantization or KV offloading, TAKE enables stable 128k-token inference on consumer-grade GPUs (24GB VRAM), where existing methods fail due to prefill-stage OOM.

**Our main contributions are summarized as follows:**

- We propose a task-aware, chunk-wise KV cache eviction framework (TAKE) for the long-context LLM inference that reduces memory usage and TTFT during the prefill stage.
- We design an accumulated task-aware probe-based mechanism to fuse task-relevant information and enable dynamic, cross-chunk token selection for KV caching.
- We present a delayed pruning strategy that mitigates shallow-layer information loss, significantly improving prefill robustness under constrained resources.
- We conduct extensive experiments on long-context benchmarks, demonstrating that TAKE outperforms prior KV eviction methods in memory efficiency and inference speed without compromising accuracy or requiring extra model retraining.

## 2 RELATED WORK

Research on KV cache eviction in long-context LLM inference encompasses methods ranging from simple sliding-window approaches to sophisticated eviction strategies. StreamingLLM (Xiao et al., 2023) retains initial and recent tokens for long-context inference, but risks discarding crucial distant information. Some approaches focus on identifying and retaining tokens with high attention importance to reduce the KV cache (Adnan et al., 2024; Dong et al., 2024). H2O (Zhang et al., 2023) preserves tokens with high accumulated attention scores over time. SnapKV (Li et al., 2024b) and MorphKV (Ghadia et al., 2025) leverage attention distributions of recent queries to identify crucial

tokens. NACL (Chen et al., 2024) augments eviction with proxy tokens and randomized selection to recover overlooked key information.

Some studies incorporate structural and semantic adaptivity into KV cache eviction. PyramidInfer (Yang et al., 2024) performs a depth-wise cache reduction, as deeper layers increasingly prioritize local over global context. GemFilter (Shi et al., 2024) and FastKV (Jo et al., 2025) allow only the hidden states of critical tokens to pass to deeper layers. CAKE (Qin et al., 2025) measures spatial dispersion and attention-shift variability to compute layer-specific preference scores for budget allocation. SepLLM (Chen et al., 2025) leverages separator tokens as summarization points for text segments, allowing it to discard intermediate tokens. ClusterKV (Liu et al., 2024b) compresses KV at the semantic level to cluster centers for efficient recall. Furthermore, the head-wise adaptive cache management methods are proposed based on the functional specialization across attention heads. Ada-KV (Feng et al., 2024) dynamically allocates per-head KV budgets via top-$k$ selection using expanded attention scores. FastGen (Ge et al., 2023) and RazorAttention (Tang et al., 2025) analyze attention modes of different heads to assign head-specific compression policies.

## 3 PRELIMINARY

Chunked prefill (Agrawal et al., 2023) partitions a long prefill into smaller chunks and interleaves them with batched decoding requests to improve GPU utilization and end-to-end throughput. Formally, the prefill phase of an input sequence is subdivided into computationally equivalent chunks of fixed size $Z$. When processing the $i$-th chunk, the KV cache is updated incrementally via concatenation, yielding a cumulative KV cache:

$$K_{i,\ell} = \text{Concat}(K_{i-1,\ell}, K_{i,\ell}^{\text{chunk}}), \tag{1}$$

$$V_{i,\ell} = \text{Concat}(V_{i-1,\ell}, V_{i,\ell}^{\text{chunk}}), \tag{2}$$

where $K_{i,\ell}$ and $V_{i,\ell}$ denote the keys and values of $i$-th chunk at layer $\ell$ used for attention computation, while $K_{i,\ell}^{\text{chunk}}$ and $V_{i,\ell}^{\text{chunk}}$ are the key and value states of the $i$-th chunk context.

Notably, all intermediate representations, including both KV and layer-wise hidden states, produced under chunked prefill are strictly identical to those from a globally full-sequence prefill. This equivalence follows from causal masking and the linearity of the projection-and-concatenation operations (see in the Appendix A for a proof). Therefore, chunked prefill is equivalent to the global prefill while enabling memory-efficient long-context inference, without compromising accuracy.

## 4 METHOD

### 4.1 OVERVIEW OF TAKE

Conventional KV cache eviction methods typically compute the full prefill before pruning to a target budget, resulting in large memory peaks and high latency. In contrast, TAKE employs a chunk-wise inference scheme during the prefill stage that alternates attention computation with KV eviction at the chunk level. This design improves GPU memory utilization and reduces prefill latency, but also introduces two main challenges: (1) Chunking weakens global coherence and causes semantic discontinuity, making it difficult to identify task-relevant critical tokens. (2) Shallow transformer layer exhibits sparse attention patterns and premature KV eviction causes substantial information loss during forward propagation.

To address these challenges, TAKE integrates two complementary mechanisms: **accumulated task-aware probe** and **delayed eviction**. The accumulated task-aware probe, a specialized token appended to each chunk, serves to identify task-relevant information and propagate semantic context across chunk boundaries. Meanwhile, delayed eviction employs a hierarchical pruning strategy at both the chunk and layer levels to enable earlier layers to benefit from more discriminative signals in deeper layers, yielding more task-aligned eviction decisions. These two mechanisms improve the accuracy of identifying important tokens and reduce information decay. It enables TAKE to avoid redundant computation without significant degradation of model performance.

Figure 2 provides an overview of the TAKE. Given an input sequence of length $N$, we partition it into $M$ non-overlapping chunks $\{C_0, C_1, \ldots, C_{M-1}\}$ with the size at most $Z$. Each chunk appends

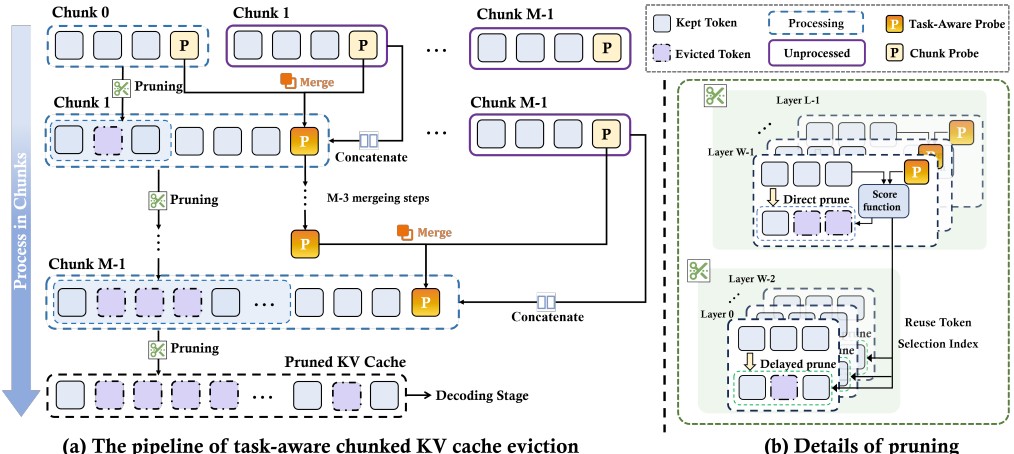

**(a) The pipeline of task-aware chunked KV cache eviction**   **(b) Details of pruning**

Figure 2: Overview of the proposed TAKE method. The left shows iterative chunk pruning with probe fusion to produce a final KV cache of size $B$. The right shows pruning within a chunk, where layer $W-1$ directs pruning for all prior layers' cache, while subsequent layers are pruned directly.

$p$ task-related probe tokens $P$, which propagate task-relevant semantics and facilitate key-token identification. Due to causal masking, probe tokens do not contribute directly to the KV cache.

We perform a complete forward propagation of $L$ decoder layers and KV cache eviction within each chunk. For the $i$-th chunk at layer $\ell$, we first merge its KV states $(K_{i,\ell}^{\mathrm{chunk}}, V_{i,\ell}^{\mathrm{chunk}})$ with the pruned summary $(\widetilde{K}_{i-1,\ell}, \widetilde{V}_{i-1,\ell})$ from the previous chunk:

$$K_{i,\ell} = \mathrm{Concat}(\widetilde{K}_{i-1,\ell}, K_{i,\ell}^{\mathrm{chunk}}), \tag{3}$$

$$V_{i,\ell} = \mathrm{Concat}(\widetilde{V}_{i-1,\ell}, V_{i,\ell}^{\mathrm{chunk}}). \tag{4}$$

Secondly, the output $O_{i,\ell}$ of the $i$-th chunk's attention operation at layer $\ell$ is computed, defined as:

$$O_{i,\ell} = \mathrm{Attention}(Q_{i,\ell}^{\mathrm{chunk}}, K_{i,\ell}, V_{i,\ell}), \tag{5}$$

where $Q_{i,\ell}^{\mathrm{chunk}}$ are the query states of $C_i$ at layer $\ell$. At the last of chunk processing, the previous chunk's task-aware probe $Q_{i-1,\ell}^{\mathrm{aprobe}}$ is merged with the current chunk's probe and input into the score function, which identifies important tokens among the current KV cache $(K_{i,\ell}, V_{i,\ell})$, as shown in Figure 2(b). A delayed eviction strategy is employed to further improve pruning quality and reduce information loss. It defers eviction for the first $W-1$ layers, and once pruning is enabled at layer $W-1$, the selected indices $\mathcal{I}_{i,W-1}$ of tokens are retrospectively applied to prune KV caches in all preceding layers. The pruning is formulated as:

$$(\widetilde{K}_{i,\ell}, \widetilde{V}_{i,\ell}) = \begin{cases} \mathrm{Prune}(\mathcal{I}_{i,W-1}, K_{i,\ell}, V_{i,\ell}), & \ell \leq W-1, \\ \mathrm{Prune}(\mathcal{I}_{i,\ell}, K_{i,\ell}, V_{i,\ell}), & \mathrm{otherwise} \end{cases}, \tag{6}$$

$$\mathcal{I}_{i,\ell} = \mathrm{Score}\Big(Q_{i,\ell}^{\mathrm{aprobe}}, K_{i,\ell}, k_{i,\ell}\Big), \tag{7}$$

$$k_{i,\ell} = \begin{cases} B_{\mathrm{warmup}}, & \ell \leq W-1 \text{ and } i \neq M-1, \\ B, & \mathrm{otherwise} \end{cases}, \tag{8}$$

where $W$ is the number of initial warm-up layers, and a relaxed KV cache budget $B_{\mathrm{warmup}}$ is applied before the last chunk to avoid excessive pruning. $\mathcal{I}_{i,\ell}$ is the selection index of top-$k_{i,\ell}$ tokens based on a score function and $k_{i,\ell}$ is the budget for KV cache of the $i$-th chunk at layer $\ell$. The pruned cache $(\widetilde{K}_{i,0\cdots L-1}, \widetilde{V}_{i,0\cdots L-1})$ is propagated forward to support attention computation in the next chunk $C_{i+1}$, where further KV eviction continues. The details of task-aware probe and delayed eviction are presented in Sec.4.2 and Sec.4.3.

Once the final chunk $C_{M-1}$ is processed, the pruned cache $(\widetilde{K}_{M-1,0\cdots L-1}, \widetilde{V}_{M-1,0\cdots L-1})$ is retained for auto-regressive decoding. Since eviction is interleaved with computation, the memory footprint at layer $\ell$ during prefill is bounded by $Z + k_{i,\ell}$ tokens, mitigating the VRAM peak.

## 4.2 ACCUMULATED TASK-AWARE PROBE

The primary risk of chunk-wise KV eviction is the disruption of global semantic consistency, as each chunk makes locally optimal decisions that conflict with the overall objective. To address this, TAKE introduces an accumulated probe mechanism, which leverages the final $p$ tokens related to the task of the input sequence as semantic anchors to guide token importance estimation across chunks. TAKE regards the KV eviction result of the previous chunk $(\widetilde{K}_{i-1,\ell}, \widetilde{V}_{i-1,\ell})$ as a summary of all

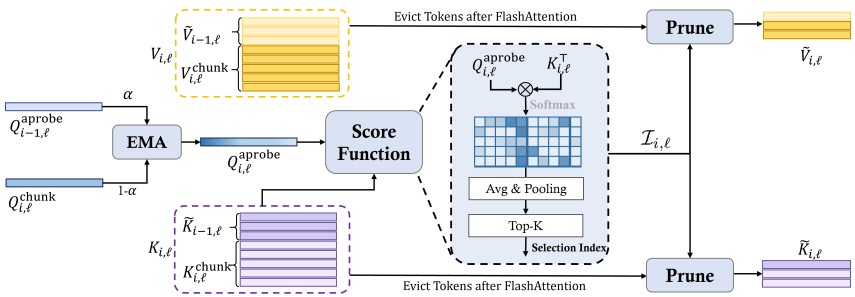

Figure 3: Probe-based KV eviction in TAKE.

preceding chunks. Since the query $Q_{i,\ell}^{probe}$ within the current chunk tends to bias toward in-chunk semantics, our proposed method fuses each chunk probe into an accumulated probe incrementally to avoid severely underestimating the importance of historical tokens. The query of the accumulated probe is updated across chunks via an exponential moving average (EMA):

$$Q_{i,\ell}^{\text{aprobe}} = \alpha\, Q_{i-1,\ell}^{\text{aprobe}} + (1-\alpha)\, Q_{i,\ell}^{\text{probe}}, \quad \alpha \in [0,1] \quad, \tag{9}$$

where $Q_{i,\ell}^{\text{aprobe}}$ is the query states of the accumulated probe to evaluate the token importance and $\alpha$ is a hyperparameter to balance semantics information between previous chunks and new chunks.

The accumulated probe acts as a dynamic, task-aware estimator that aligns with global objectives under chunked eviction. Detailed illustration of KV eviction based on the probe is shown in Figure 3. TAKE computes the matrix inner product between $Q_{i,\ell}^{\text{aprobe}} \in \mathbb{R}^{b \times h \times p \times d}$ and $K_{i,\ell}^{T} \in \mathbb{R}^{b \times h \times d \times (Z+k_{i-1,\ell})}$ and then applies softmax, producing a tensor $A_{i,\ell} \in \mathbb{R}^{b \times h \times p \times (Z+k_{i-1,\ell})}$, where $b$ denotes batch size, $h$ denotes the number of attention heads, and $d$ is the hidden state dimension. We average $A_{i,\ell}$ along the probe dimension of length $p$, and then apply mean pooling over the last dimension to form token importance scores $S_{i,\ell}$ of the current KV cache. Finally, the collection of selected token index $\mathcal{I}_{i,\ell}$ is obtained through sorting $S_{i,\ell}$, based on which $(\widetilde{K}_{i,\ell}, \widetilde{V}_{i,\ell})$ are gathered from $(K_{i,\ell}, V_{i,\ell})$. We formalize the above KV eviction procedure as the following equations.

$$A_{i,\ell} = \text{Softmax}\left(\frac{Q_{i,\ell}^{\text{aprobe}}(K_{i,\ell})^{\top}}{\sqrt{d}}\right) \in \mathbb{R}^{b \times h \times p \times (Z+k_{i-1,\ell})}, \tag{10}$$

$$S_{i,\ell} = \text{AvgPool}\left(\frac{1}{p}\sum_{u=1}^{p} A_{i,\ell}[:,:,u,:];\ \text{kernel} = r, \dim = 2\right) \in \mathbb{R}^{b \times h \times (Z+k_{i-1,\ell})}, \tag{11}$$

$$\mathcal{I}_{i,\ell} = \text{Top-}k_{i,\ell}(S_{i,\ell};\ \dim = 2), \tag{12}$$

$$\widetilde{K}_{i,\ell} = \text{Gather}(K_{i,\ell}, \mathcal{I}_{i,\ell}), \quad \widetilde{V}_{i,\ell} = \text{Gather}(V_{i,\ell}, \mathcal{I}_{i,\ell}). \tag{13}$$

To preserve the functional diversity of multi-head attention, there is no aggregation applied across the head dimension. The KV pairs corresponding to the top-$k_{i,\ell}$ tokens are retained as $(\widetilde{K}_{i,\ell}, \widetilde{V}_{i,\ell})$ in GPU memory to participate in the prefilling of the next chunk.

## 4.3 DELAYED EVICTION

The effectiveness of KV eviction methods depends on attention sparsity. However, the Figure 4 reveals that the attention distribution of early transformer layers lacks sufficient sparsity for reliable pruning. For example, in the NIAH task, Layer 0 assigns nearly uniform attention across the sequence, while deeper layers (e.g., Layer 15) clearly highlight the target token.

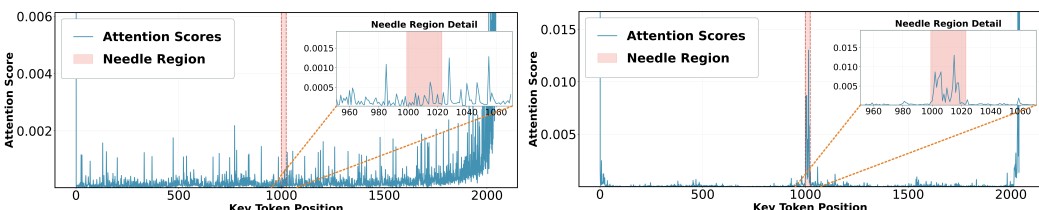

(a) Attention scores of last token at Layer 0.    (b) Attention scores of last token at Layer 15.

Figure 4: Attention scores of different layers on Needle-in-a-Haystack benchmark.

Therefore, early pruning in shallow layers can induce distortions in attention distributions as a consequence of renormalization, thereby deteriorating hidden state representations and language modeling. This challenge is dominant in chunk-wise KV eviction scenarios, where each chunk attends exclusively to summarized representations of preceding chunks rather than the entire prior context. To address this limitation, the TAKE framework implements a delayed KV eviction strategy along two distinct dimensions: the chunk-level delayed pruning and the layer-level delayed pruning.

**Chunk-level delayed pruning** For the initial $W$ shallow layers, a relaxed budget $B_{\text{warmup}}$ is applied, while deeper layers are pruned to the final target budget $B$ due to attention sparsity. At the last chunk, shallow-layer caches are reduced from $B_{\text{warmup}}$ to the target size $B$. Chunk-level delayed pruning makes shallow layers to integrate cross-chunk evidence before eviction, yielding more semantic representations while ensuring decoding starts with a consistent cache size.

**Layer-level delayed pruning** Since shallow layers mainly handle syntactic and semantic parsing rather than token discrimination selection, TAKE synchronizes pruning decisions from a deeper key layer. Specifically, when pruning is first enabled at layer $W$, the token indices selected at this layer are used to update the KV caches of all preceding layers simultaneously. This key-layer decision, global synchronization strategy ensures that shallow layer caches inherit reliable importance decisions without requiring separate scoring and selection, thereby reducing computation overhead and preserving early semantic features with task-relevant selections.

## 5 EXPERIMENTS

### 5.1 EXPERIMENTAL SETTINGS

**LLMs & Evaluation Tasks** Experiments were conducted on two open-source LLMs with different sizes: Llama-3.1-8B-Instruct (Grattafiori et al., 2024) and Mistral-7B-Instruct-v0.3 (Jiang et al., 2023). We evaluate our method on two long-context benchmarks: NIAH (Kamradt, 2023) and LongBench (Bai et al., 2023). NIAH embeds sparse key information into long and distracting texts. LongBench is a multitask suite for long-context evaluation, covering extended-context QA, summarization, few-shot learning, and coding tasks.

**Baseline Methods** We compare TAKE against Full-KV and four representative KV cache eviction strategies: SnapKV (Li et al., 2024b), AdaKV (Feng et al., 2024), FastKV (Jo et al., 2025) and CAKE (Qin et al., 2025). Full-KV serves as the baseline without pruning. SnapKV represents importance-based token retention. AdaKV implements adaptive per-head KV budgets. FastKV applies pruning both on hidden states and KV cache.

**Implementation Details** All experiments are conducted on NVIDIA H800 GPUs under uniform hardware conditions. TAKE is built upon the HuggingFace Transformers library, ensuring compati-

bility with both GQA (Ainslie et al., 2023) and FlashAttention-2 (Dao, 2024) backends. For hyper-parameter settings, we apply delayed pruning to the initial 50% of the model layers and $B_{\mathrm{warmup}}$ is decided by context length. The EMA factor $\alpha$ in the accumulated probe is set to 0.32 for NIAH and 0.2 respectively for LongBench based on task properties.

## 5.2 EVALUATION ON NEEDLE-IN-A-HAYSTACK AND LONGBENCH

**Needle-in-a-Haystack**   As shown in Figure 5, we compare TAKE to baseline methods on the NIAH benchmark based on LLaMA-3.1-8B-Instruct with $512L$ KV budget. The green denotes 100% accuracy while red denotes 0%. The results show that TAKE consistently outperforms prior eviction strategies on NIAH, achieving stronger localization and recall of sparse information. In Appendix C.1, additional comparison results on LLaMA-3.1-8B-Instruct and Mistral-7B-Instruct-v0.3 with $256L$, $512L$ and $1024L$ KV budgets are presented. Interestingly, TAKE even surpasses standard full-KV inference by filtering irrelevant content during prefill. These results demonstrate that reducing the KV cache size improves LLM inference performance when processing input sequences with excessive redundant information, further highlighting the effectiveness of task-aware pruning.

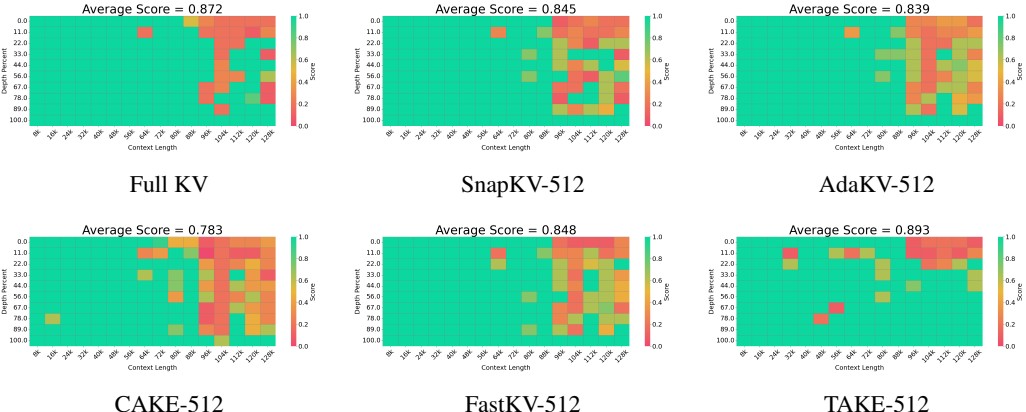

Figure 5: Needle-in-a-Haystack results of LLaMA-3.1-8B-Instruct with $512L$ KV budget.

**LongBench**   Table 1 reports results of LongBench on LLaMA-3.1-8B-Instruct and Mistral-7B-Instruct-v0.3 with $128L$ and $512L$ KV budgets. With a KV budget of $512L$ on LLama-3.1-8B-Instruct, TAKE achieves an average score of 47.45, outperforming SnapKV by +0.98, AdaKV by +1.24, FastKV by +0.79, and CAKE by +1.08. TAKE consistently outperforms baseline pruning methods under limited KV budgets (see more results in Appendix C.2), especially in multi-document question answering and coding tasks, where the effective integration of long-range dependencies and retention of key tokens are crucial. It greatly demonstrates the robustness and advancement of TAKE across two LLMs with various KV budgets.

## 5.3 EVALUATION ON MEMORY USAGE AND TTFT

The memory usage and time-to-fist-token(TTFT) are evaluated on an NVIDIA H800 GPU by averaging results over multiple runs. Figure 6(a) shows that for 128k-token inference, full-KV and other global KV eviction methods approximately demand around 48.91GB and 31.90GB of GPU memory. By contrast, TAKE only requires 18.99GB including the 16.07GB allocated for model weights, due to its chunked prefill design, limiting the peak memory usage. This reduction translates to a 91.11% decrease in memory of KV cache and activation compared to full-KV inference, and an 81.55% decrease compared to global KV eviction baselines.

In terms of inference speed, TAKE also demonstrates substantial gains. Global pruning methods retain full-sequence attention before pruning, resulting in TTFT that matches or exceeds native full-KV due to added overhead. FastKV reduces hidden-state size but still computes attention over the entire sequence in shallow layers. As illustrated in Figure 6(b), TAKE cuts prefill latency by over 60% compared to the baseline and by 30.9% compared to FastKV.

Table 1: LongBench results of LLaMA-3.1-8B-Instruct and Mistral-7B-Instruct-v0.3 with $128L$ and $512L$ KV budgets.

| Method | Single-Doc QA | | | Multi-Doc QA | | | Summarization | | | Few-shot Learning | | | Coding | | Avg. |
|---|---|---|---|---|---|---|---|---|---|---|---|---|---|---|---|
| | NrtvQA | Qasper | MF-en | HotpotQA | 2WikiMQA | Musique | GovRep. | QMSum | MultiNews | TREC | TriviaQA | SAMSum | LCC | RB-P | |
| **LLaMA-3.1-8B-Instruct**, *KV Budget = Full* | | | | | | | | | | | | | | | |
| Full KV | 30.21 | 45.53 | 55.01 | 56.01 | 46.65 | 31.28 | 35.13 | 25.28 | 27.25 | 73.00 | 91.64 | 43.80 | 63.38 | 56.64 | 48.63 |
| **LLaMA-3.1-8B-Instruct**, *KV Budget = 128L* | | | | | | | | | | | | | | | |
| SnapKV | 27.23 | 30.96 | 50.62 | 53.04 | 43.13 | 28.66 | 22.70 | 23.16 | 21.90 | 62.00 | 91.19 | 40.37 | 58.37 | 49.09 | 43.03 |
| AdaKV | 27.71 | 30.13 | 50.27 | 51.65 | 40.87 | 29.21 | 23.42 | 23.08 | 21.72 | 67.50 | 91.10 | 40.54 | 58.87 | 50.15 | 43.30 |
| FastKV | 27.49 | 28.05 | 51.56 | 52.12 | 44.54 | 28.78 | 22.12 | 23.04 | 20.77 | 59.50 | 91.06 | 40.79 | 59.03 | 49.16 | 42.71 |
| CAKE | 27.41 | 32.01 | 49.58 | 52.99 | 45.76 | 28.32 | 21.85 | 23.15 | 21.56 | 47.50 | 89.61 | 40.90 | 57.92 | 49.83 | 42.03 |
| TAKE | 27.39 | 35.36 | 51.46 | 52.94 | 46.69 | 29.47 | 22.00 | 22.25 | 26.19 | 59.00 | 91.29 | 39.06 | 61.33 | 52.76 | 44.09 |
| **LLaMA-3.1-8B-Instruct**, *KV Budget = 512L* | | | | | | | | | | | | | | | |
| SnapKV | 30.04 | 41.14 | 53.83 | 54.77 | 45.12 | 31.20 | 27.08 | 24.04 | 25.00 | 70.50 | 91.90 | 42.34 | 61.34 | 52.24 | 46.47 |
| AdaKV | 30.09 | 40.20 | 54.14 | 54.27 | 44.73 | 30.77 | 26.84 | 23.98 | 24.17 | 70.50 | 91.90 | 41.74 | 60.63 | 53.02 | 46.21 |
| FastKV | 29.96 | 40.88 | 54.55 | 54.78 | 46.19 | 30.76 | 26.63 | 24.05 | 24.29 | 72.50 | 91.37 | 42.52 | 62.14 | 52.65 | 46.66 |
| CAKE | 31.82 | 42.99 | 51.65 | 54.37 | 46.89 | 30.73 | 26.36 | 24.94 | 25.27 | 63.50 | 91.54 | 42.52 | 62.31 | 54.30 | 46.37 |
| TAKE | 30.45 | 43.22 | 55.20 | 56.99 | 46.77 | 30.18 | 26.44 | 23.77 | 26.60 | 70.50 | 92.48 | 41.84 | 63.25 | 56.67 | 47.45 |
| **Mistral-7B-Instruct-v0.3**, *KV Budget = Full* | | | | | | | | | | | | | | | |
| Full KV | 29.20 | 38.27 | 50.09 | 51.20 | 36.49 | 26.67 | 34.21 | 25.79 | 26.43 | 76.00 | 88.59 | 47.46 | 59.37 | 60.52 | 46.45 |
| **Mistral-7B-Instruct-v0.3**, *KV Budget = 128L* | | | | | | | | | | | | | | | |
| SnapKV | 23.77 | 28.60 | 45.64 | 44.60 | 29.85 | 21.92 | 22.66 | 22.56 | 21.32 | 47.50 | 87.94 | 43.32 | 54.67 | 51.75 | 39.00 |
| AdaKV | 24.54 | 29.42 | 45.65 | 42.40 | 28.66 | 22.18 | 22.89 | 23.36 | 21.53 | 65.50 | 88.89 | 44.55 | 53.89 | 53.23 | 40.18 |
| FastKV | 24.97 | 26.68 | 45.46 | 43.75 | 30.01 | 21.06 | 22.43 | 22.33 | 20.37 | 63.50 | 89.24 | 42.50 | 54.37 | 51.80 | 39.89 |
| CAKE | 22.31 | 29.15 | 43.51 | 44.51 | 30.36 | 22.85 | 21.56 | 20.47 | 18.96 | 47.00 | 88.60 | 39.36 | 44.96 | 46.19 | 37.13 |
| TAKE | 23.25 | 28.56 | 43.64 | 48.80 | 34.32 | 20.92 | 22.06 | 21.86 | 25.63 | 56.50 | 89.16 | 43.19 | 58.40 | 51.97 | 40.43 |
| **Mistral-7B-Instruct-v0.3**, *KV Budget = 512L* | | | | | | | | | | | | | | | |
| SnapKV | 26.03 | 34.82 | 49.74 | 49.25 | 34.11 | 23.88 | 27.23 | 24.33 | 24.59 | 74.00 | 89.82 | 45.94 | 58.13 | 58.49 | 44.31 |
| AdaKV | 27.14 | 35.10 | 48.65 | 48.98 | 34.30 | 23.80 | 25.76 | 24.38 | 24.27 | 75.00 | 89.14 | 45.87 | 58.18 | 59.49 | 44.29 |
| FastKV | 25.94 | 33.20 | 49.03 | 48.37 | 34.42 | 23.43 | 26.63 | 23.88 | 24.10 | 71.50 | 89.27 | 46.49 | 58.72 | 58.63 | 43.97 |
| CAKE | 25.82 | 34.24 | 46.93 | 45.34 | 31.71 | 23.38 | 25.81 | 21.47 | 22.41 | 68.00 | 88.46 | 41.27 | 49.04 | 49.67 | 40.96 |
| TAKE | 26.65 | 35.61 | 49.49 | 50.11 | 37.42 | 22.99 | 26.18 | 24.09 | 25.98 | 73.50 | 89.54 | 45.12 | 59.86 | 58.18 | 44.62 |

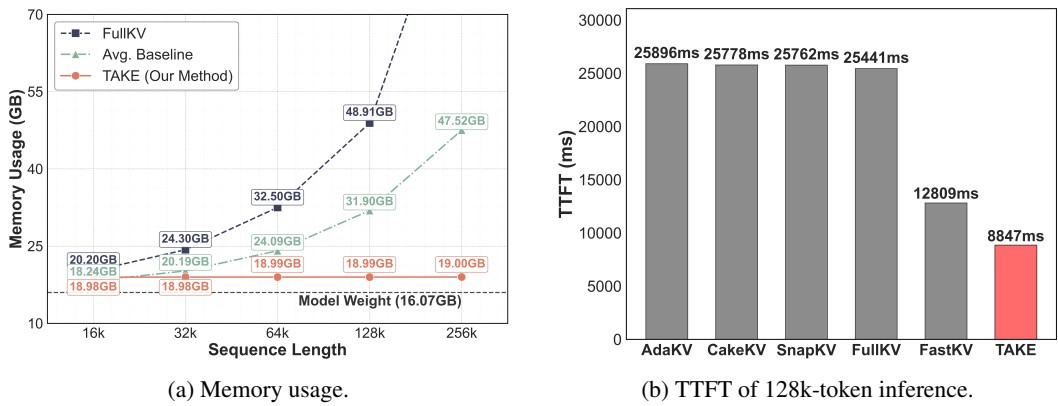

(a) Memory usage.  (b) TTFT of 128k-token inference.

Figure 6: Memory usage and TTFT comparisons for different KV pruning methods.

These memory and latency improvements arise from chunked eviction, which eliminates excessive padding and confines attention in subsequent chunks to pruned summaries rather than the full prior context, yielding substantial computational savings. We further estimate floating-point operations (FLOPs) in Appendix B. For 128k-token inference, TAKE requires only 7.1% of the FLOPs of full-KV. Although chunking introduces extra kernel launches and memory operations, the overall reduction in computation outweighs these overheads, resulting in a significant decrease in TTFT.

## 5.4 ABLATION STUDY

**Effect of task-aware probe and delayed eviction** As shown in Table 2, we evaluate the effectiveness of the accumulated task-aware probe and delayed eviction on LLama-3.1-8B under the $512L$

KV budget. There are four baseline settings: only chunk-wise eviction (chunk-only), chunk-wise delayed eviction without probe (w/o probe), chunk-wise eviction with probe (w/o delay), and chunk-wise delayed eviction with probe (TAKE). The accuracy of w/o delayed eviction on the NIAH with a 64k context length reaches 80.00%, a 69.10% gain over the chunk-only methods. On LongBench datasets, the w/o delayed eviction method achieves a 19.43 improvement compared with chunk-only eviction. The results of chunk-wise delayed eviction without probe are the same as the results with only chunk-wise eviction on the two benchmarks. It confirms the inefficacy of delayed eviction when used in isolation and the cooperative efficacy of the two strategies in the TAKE.

Table 2: Ablation study on LLama-3.1-8B under $512L$ KV Budget (w/o means without).

| Configuration | Chunk-wise | Probe | Delay | NIAH Accuracy on 64k-token | | LongBench Macro Avg. | |
|---|---|---|---|---|---|---|---|
| | | | | Value | $\Delta$ vs Chunk-only | Value | $\Delta$ vs Chunk-only |
| Chunk-only | ✓ | ✗ | ✗ | 10.90% | – | 26.46 | – |
| w/o Probe | ✓ | ✗ | ✓ | 10.90% | +00.00% | 26.46 | +00.00 |
| w/o Delay | ✓ | ✓ | ✗ | 80.00% | +69.10% | 45.89 | +19.43 |
| **TAKE (Ours)** | ✓ | ✓ | ✓ | **91.8%** | **+80.90%** | **47.45** | **+20.99** |

**Effect of warm-up parameters** The number of warm-up layers, $W$, and the warm-up budget, $B_{\mathrm{warmup}}$ are critical in the delayed pruning. We perform analysis of $W$ and $B_{\mathrm{warmup}}$ on NIAH and LongBench on LLama-3.1-8B with $512L$ budget shown in Figure 7(a) and Figure 7(b). For the NIAH passkey retrieval task, reducing $W$ improves accuracy, as it allows the model to start filtering out irrelevant information at an earlier stage. For warm-up budget, increasing $B_{\mathrm{warmup}}$ improves performance across both benchmarks by retaining more information in shallow layers. This performance gain introduces more TTFT and GPU memory consumption, highlighting a clear trade-off between accuracy and computation cost.

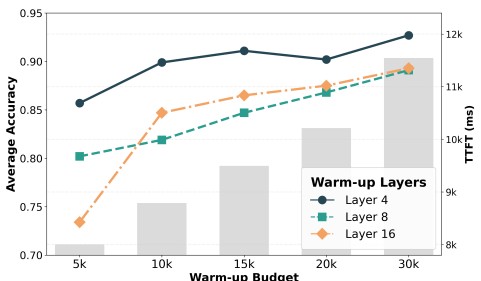

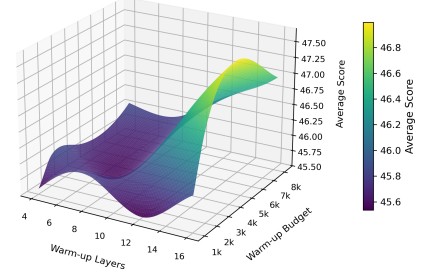

(a) Average scores (line) on NIAH across various warm-up parameters and TTFT (bar) of 128k-token inference with 16 warm-up layers.

(b) Average scores on Longbench across various warm-up parameters.

Figure 7: Analysis of warm-up parameters.

## 6    CONCLUSION

In this paper, we present TAKE, a training-free, chunk-wise KV cache eviction method that balances memory efficiency and model performance for efficient long-context LLMs inference. Unlike prior methods that prune the KV cache after full attention computation, TAKE interleaves computation with pruning to reduce the peak GPU memory usage. Extensive experiments on two typical benchmarks demonstrate that TAKE achieves superior memory savings and inference latency, enabling stable inference with arbitrary sequence length.

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

# A    PROOF OF EQUIVALENCE BETWEEN CHUNKED PREFILL AND ORIGINAL PREFILL

Let $X = (x_0, \ldots, x_{N-1})$ be an input sequence of $N$ tokens, and denote $h_j^{(l)}$ as the hidden state of token $x_j$ at layer $\ell$. The base case at layer $\ell = 0$ corresponds to the token embedding layer, where both chunked prefill and original prefill receive identical raw input embeddings, thus $h_j^{(0)}$ are the same.

Assuming as induction hypothesis that for layer $l - 1$, both methods produce identical hidden states $h_j^{(l-1)}$ for all tokens, we prove that this holds true at layer $l$ as well. In layer $l$, the query, key and value vectors $(q_j, k_j, v_j)$ of the token $x_j$ are computed using linear projections in $h_j^{(l-1)}$ with weight matrices $W_Q, W_K, W_V$. By the hypothesis, $h_j^{(l-1)}$ are the same for both methods, so the projections yield identical $q_j, k_j, v_j$.

The self-attention output $O_j = \text{Attention}(q_j, K_{1\ldots j}, V_{1\ldots j})$ attends to token $x_j$ and all preceding tokens, respecting the causal mask. In the global prefill mode, $K$ and $V$ are computed once for the full sequence; in chunked prefill, the previously computed KV caches store all preceding blocks' keys and values. The attention context for $x_j$ in either mode is the same: $K_{1..j}$ and $V_{1..j}$. Thus, outputs $O_j$ are identical. This equality relies on the linearity of projection and the additive property of concatenated KV vectors, ensuring perfect correspondence between chunked and full-sequence generated KV states.

Since attention outputs $O_j$ and inputs $h_j^{(l-1)}$ are identical, subsequent operations within the layer, such as residual connections, layer normalization, and position-wise feedforward network, also produce identical outputs, leading to exactly matching hidden states $h_j^{(l)}$.

By induction, chunked prefill and full prefill generate perfectly equivalent hidden states and KV caches across all layers, with no information loss.

# B    FLOPS CALCULATION

**Full-KV Method**    In the full-KV attention method without chunking, the query, key, and value tensors are $Q, K, V \in \mathbb{R}^{b \times H \times N \times d}$, where $b$ is batch size, $H$ is number of attention heads, $N$ is sequence length, and $d$ is hidden dimension. The attention operation is

$$Attention(Q, K, V) = \text{softmax}\left(\frac{QK^T}{\sqrt{d}}\right)V. \tag{14}$$

The floating-point operations (FLOPs) consist of

- $2bHN^2d$ for $QK^T$ matrix multiplication,

- $bHN^2$ for scaling by $\frac{1}{\sqrt{d}}$,

- $3bHN^2$ for the softmax normalization,

- $2bHN^2d$ for multiplication with $V$.

Summing these yields the total FLOPs for $L$ layers:

$$FLOPs_{\text{FullKV}} = 4bLHN^2(d+1). \tag{15}$$

**TAKE Method**    TAKE partitions the input into $M$ chunks each of size $Z$, with pruning budget $B_{\text{warmup}}$ for half layers of model and $B$ for remaining layers. Length of task-aware probe is $p$. The FLOPs include chunked attention and token scoring overhead:

Chunked attention FLOPs:

$$
\begin{aligned}
FLOPs_{\text{TAKE-attention}} = 4b\frac{L}{2}H\big[ MZ^2(d+1) \\
+(M-1)(Z+B_{\text{warmup}})(Z+p)(d+1)\big] \\
+4b\frac{L}{2}H\big[ MZ^2(d+1) \\
+(M-1)(Z+B)(Z+p)(d+1)\big].
\end{aligned}
\tag{16}
$$

Token scoring FLOPs (performed on half the layers due to delayed pruning):

$$
FLOPs_{\text{TAKE-prune}} = \frac{L}{2}\big[ 2bHpZ(d+1) + 2bHp(M-1)(Z+B)(d+1)\big].
\tag{17}
$$

The total FLOPs for TAKE is:

$$
\begin{aligned}
FLOPs_{\text{TAKE}} &= FLOPs_{\text{TAKE-attention}} + FLOPs_{\text{TAKE-prune}} \\
&= bLH(d+1)\bigg[ 4MZ^2 + 5MZp \\
&\quad + \frac{5p(M-1)(B+B_{\text{warmup}})}{2} \\
&\quad + 2Z(M-1)(B+B_{\text{warmup}})\bigg]
\end{aligned}
\tag{18}
$$

The FLOPs ratio is calculated by:

$$
\begin{aligned}
r &= \frac{FLOPs_{\text{TAKE}}}{FLOPs_{\text{FullKV}}} \\
&= \frac{1}{8N^2}\big( 8MZ^2 + 10MZp \\
&\quad + 5p(M-1)(B+B_{\text{warmup}}) \\
&\quad + 4Z(M-1)(B+B_{\text{warmup}})\big).
\end{aligned}
\tag{19}
$$

Given $N \approx MZ$ and $p \ll N$, simplifying yields:

$$
\begin{aligned}
r &\approx \frac{Z}{N} + \frac{5p}{4N} + \frac{5p(B+B_{\text{warmup}})}{8ZN} + \frac{B+B_{\text{warmup}}}{2N} \\
&\quad - \frac{5p(B+B_{\text{warmup}})}{8N^2} - \frac{Z(B+B_{\text{warmup}})}{2N^2} \\
&\approx \frac{2Z + B + B_{\text{warmup}}}{2N} - \frac{Z(B+B_{\text{warmup}})}{2N^2}.
\end{aligned}
\tag{20}
$$

Substituting experimental parameters $Z = 4096$, $B_{\text{warmup}} = 10240$, $B = 512$, $N = 128000$, we obtain

$$
r \approx 7.10\%.
$$

Thus, TAKE reduces computation to roughly 7.10% of the full-KV baseline, showing significant efficiency gains.

## C  ADDITIONAL EXPERIMENTAL RESULTS

### C.1  NEEDLE-IN-A-HAYSTACK

We report NIAH results on LLaMa-3.1-8B-Instruct and Mistral-7B-Instruct-v0.3 under different KV budgets. Proper KV size takes positive effect on retrieval accuracy while larger ones may introduce noise.

#### C.1.1  NEEDLE-IN-A-HAYSTACK ON LLAMA-3.1-8B-INSTRUCT

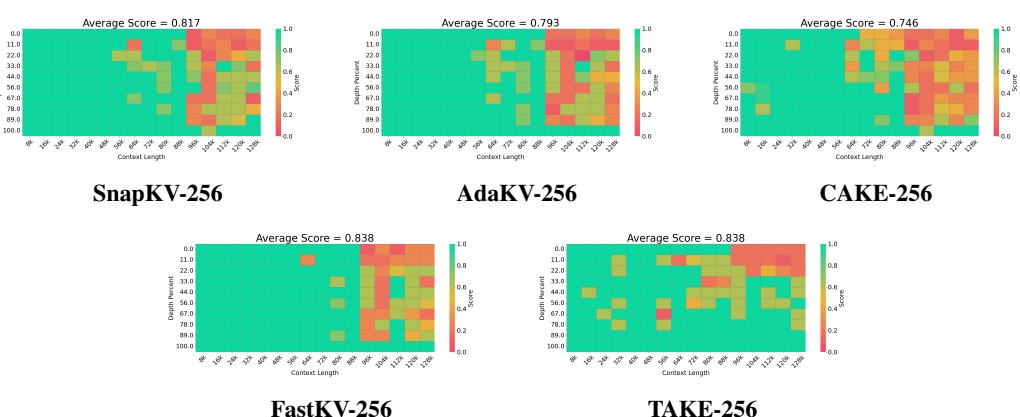

Figure 8: Needle-in-a-Haystack results of LLaMA-3.1-8B-Instruct with $256L$ KV budget.

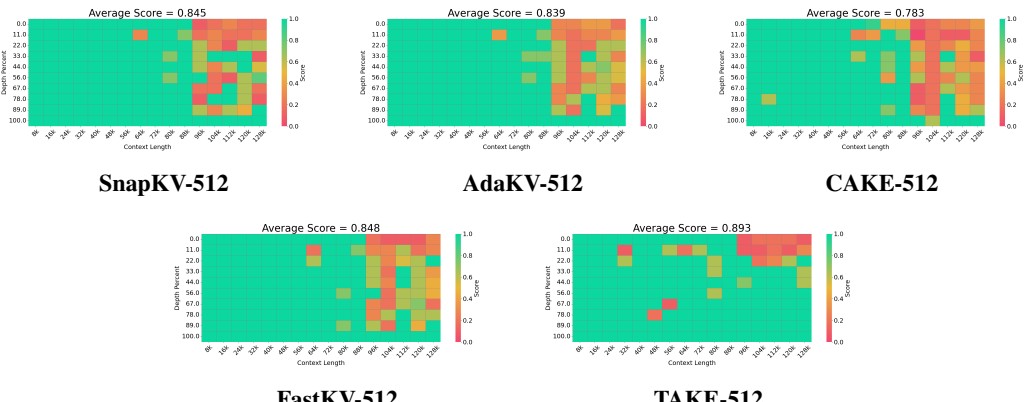

Figure 9: Needle-in-a-Haystack results of LLaMA-3.1-8B-Instruct with $512L$ KV budget.

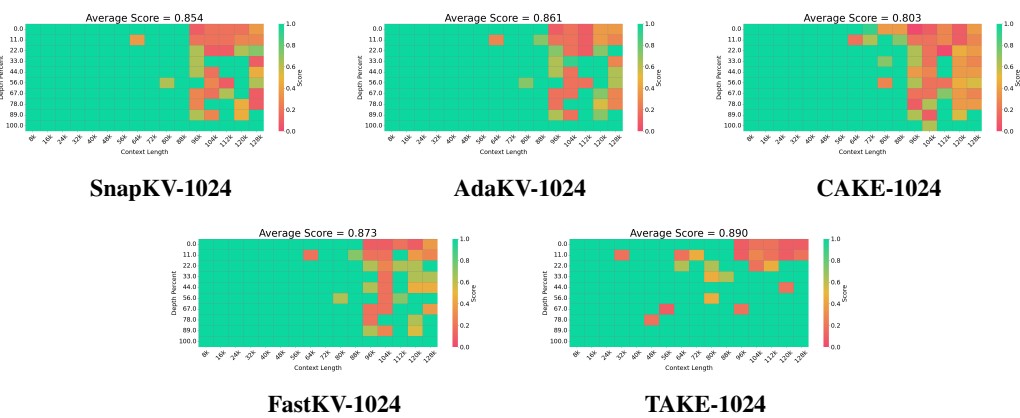

Figure 10: Needle-in-a-Haystack results of LLaMA-3.1-8B-Instruct with $1024L$ KV budget.

### C.1.2 NEEDLE-IN-A-HAYSTACK ON MISTRAL-7B-INSTRUCT-V0.3

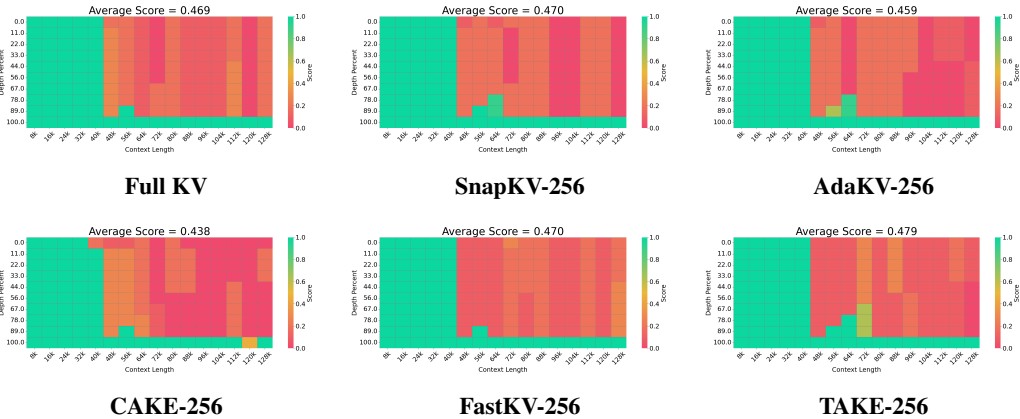

Figure 11: Needle-in-a-Haystack results of Mistral-7B-Instruct-v0.3 with 256$L$ KV budget.

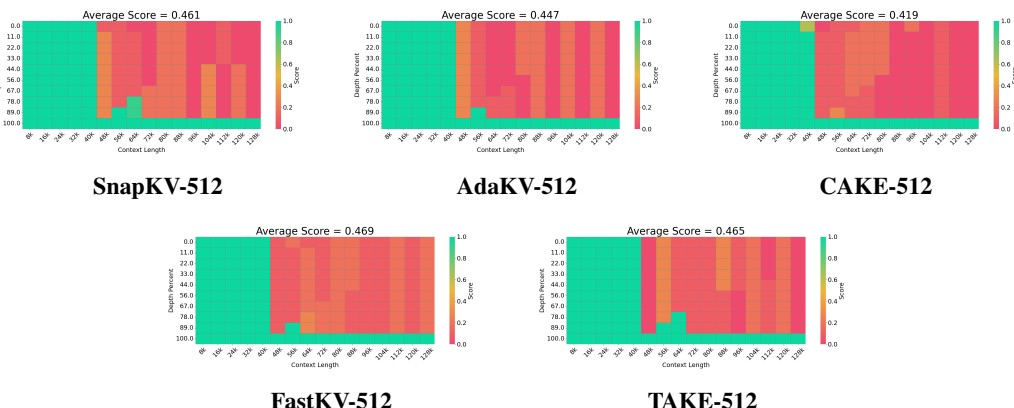

Figure 12: Needle-in-a-Haystack results of Mistral-7B-Instruct-v0.3 with 512$L$ KV budget.

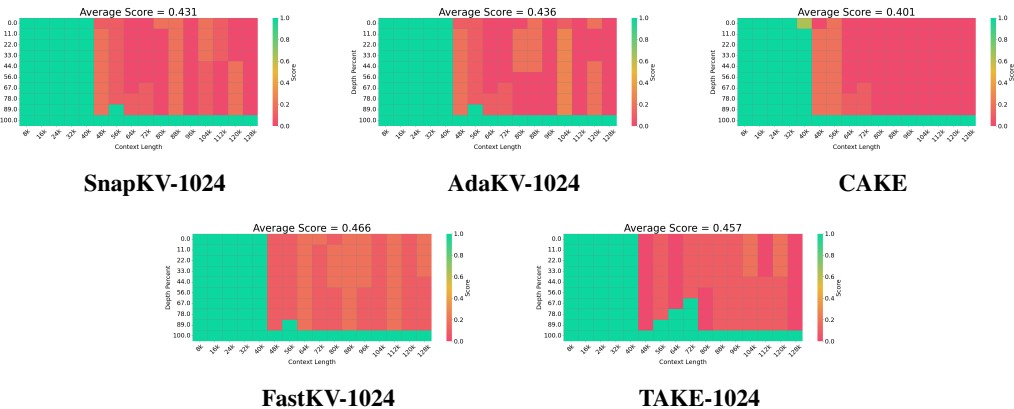

Figure 13: Needle-in-a-Haystack results of Mistral-7B-Instruct-v0.3 with 1024$L$ KV budget.

### C.2 LONGBENCH

We provide detailed LongBench results under different KV budgets for LLaMa-3.1-8B-Instruct (Table 3) and Mistral-7B-Instruct-v0.3 (Table 4).

Table 3: LongBench results comparison on LLaMA-3.1-8B-Instruct

| Method | Single-Doc QA | | | Multi-Doc QA | | | Summarization | | | Few-shot Learning | | | Coding | | Avg. |
|---|---|---|---|---|---|---|---|---|---|---|---|---|---|---|---|
| | NrtvQA | Qasper | MF-en | HotpotQA | 2WikiMQA | Musique | GovRep | QMSum | MultiNews | TREC | TriviaQA | SAMSum | LCC | RB-P | |
| **LLaMA-3.1-8B-Instruct**, *KV Budget = Full* | | | | | | | | | | | | | | | |
| Full KV | 30.25 | 45.53 | 55.01 | 56.01 | 46.65 | 31.28 | 35.13 | 25.28 | 27.25 | 73.00 | 91.64 | 43.80 | 63.38 | 56.64 | 48.63 |
| **LLaMA-3.1-8B-Instruct**, *KV Budget = 128* | | | | | | | | | | | | | | | |
| SnapKV | 27.23 | 30.96 | 50.62 | 53.04 | 43.13 | 28.66 | 22.70 | 23.16 | 21.90 | 62.00 | 91.19 | 40.37 | 58.37 | 49.09 | 43.03 |
| AdaKV | 27.71 | 30.13 | 50.27 | 51.65 | 40.87 | 29.21 | 23.42 | 23.08 | 21.72 | 67.50 | 91.10 | 40.54 | 58.87 | 50.15 | 43.30 |
| FastKV | 27.49 | 28.05 | 51.56 | 52.12 | 44.54 | 28.78 | 22.12 | 23.04 | 20.77 | 59.50 | 91.06 | 40.79 | 59.03 | 49.16 | 42.71 |
| CAKE | 27.41 | 32.01 | 49.58 | 52.99 | 45.76 | 28.32 | 21.85 | 23.15 | 21.56 | 47.50 | 89.61 | 40.90 | 57.92 | 49.83 | 42.03 |
| TAKE | 27.39 | 35.36 | 51.46 | 52.94 | 46.69 | 29.47 | 22.00 | 22.25 | 26.19 | 59.00 | 91.29 | 39.06 | 61.33 | 52.76 | 44.09 |
| **LLaMA-3.1-8B-Instruct**, *KV Budget = 256L* | | | | | | | | | | | | | | | |
| SnapKV | 29.51 | 36.43 | 52.75 | 54.06 | 43.38 | 28.80 | 25.21 | 23.98 | 23.40 | 69.00 | 91.96 | 41.26 | 60.16 | 50.31 | 45.02 |
| AdaKV | 29.17 | 34.41 | 51.67 | 54.53 | 42.79 | 30.54 | 25.09 | 23.56 | 23.19 | 70.50 | 92.12 | 41.15 | 59.80 | 51.16 | 44.98 |
| FastKV | 29.70 | 35.74 | 53.85 | 54.45 | 45.54 | 29.95 | 24.59 | 23.63 | 23.10 | 67.00 | 91.55 | 41.59 | 60.98 | 50.52 | 45.16 |
| CAKE | 29.75 | 38.74 | 52.27 | 54.52 | 45.39 | 29.57 | 24.15 | 24.24 | 23.77 | 57.00 | 90.97 | 41.82 | 61.01 | 51.90 | 44.65 |
| TAKE | 29.36 | 39.85 | 53.84 | 55.01 | 44.65 | 25.25 | 24.03 | 22.92 | 26.31 | 63.50 | 91.81 | 41.42 | 62.03 | 54.44 | 45.60 |
| **LLaMA-3.1-8B-Instruct**, *KV Budget = 512L* | | | | | | | | | | | | | | | |
| SnapKV | 30.04 | 41.14 | 53.83 | 54.77 | 45.12 | 31.20 | 27.08 | 24.04 | 25.00 | 70.50 | 91.90 | 42.34 | 61.34 | 52.24 | 46.47 |
| AdaKV | 30.09 | 40.20 | 54.14 | 54.27 | 44.73 | 30.77 | 26.84 | 23.98 | 24.17 | 70.50 | 91.90 | 41.74 | 60.63 | 53.02 | 46.21 |
| FastKV | 29.96 | 40.88 | 54.55 | 54.78 | 46.19 | 30.76 | 26.63 | 24.05 | 24.29 | 72.50 | 91.37 | 42.52 | 62.14 | 52.65 | 46.66 |
| CAKE | 31.82 | 42.99 | 51.65 | 54.37 | 46.89 | 30.73 | 26.36 | 24.94 | 25.27 | 63.50 | 91.54 | 42.52 | 62.31 | 54.30 | 46.37 |
| TAKE | 30.45 | 43.22 | 55.20 | 56.99 | 46.77 | 30.18 | 26.44 | 23.77 | 26.60 | 70.50 | 92.48 | 41.84 | 63.25 | 56.67 | 47.45 |
| **LLaMA-3.1-8B-Instruct**, *KV Budget = 1024L* | | | | | | | | | | | | | | | |
| SnapKV | 31.23 | 42.52 | 53.96 | 55.48 | 45.43 | 31.50 | 29.54 | 24.78 | 26.17 | 70.50 | 91.73 | 42.52 | 62.25 | 54.72 | 47.31 |
| AdaKV | 31.00 | 44.35 | 54.58 | 55.86 | 45.39 | 31.96 | 28.77 | 24.92 | 26.04 | 71.00 | 91.48 | 42.32 | 61.88 | 54.75 | 47.45 |
| FastKV | 30.21 | 43.86 | 55.16 | 55.14 | 46.06 | 30.65 | 29.42 | 24.18 | 26.14 | 73.50 | 91.23 | 43.42 | 62.63 | 55.69 | 47.66 |
| CAKE | 30.88 | 44.95 | 52.38 | 55.49 | 46.99 | 30.82 | 28.68 | 24.91 | 26.39 | 69.00 | 91.94 | 42.60 | 62.65 | 56.89 | 47.47 |
| TAKE | 30.83 | 44.53 | 54.37 | 56.42 | 47.30 | 27.31 | 28.78 | 24.42 | 26.79 | 72.00 | 92.41 | 40.92 | 63.32 | 55.90 | 47.52 |

Table 4: LongBench results comparison on Mistral-7B-Instruct-v0.3

| Method | Single-Doc QA | | | Multi-Doc QA | | | Summarization | | | Few-shot Learning | | | Coding | | Avg. |
|---|---|---|---|---|---|---|---|---|---|---|---|---|---|---|---|
| | NrtvQA | Qasper | MF-en | HotpotQA | 2WikiMQA | Musique | GovRep | QMSum | MultiNews | TREC | TriviaQA | SAMSum | LCC | RB-P | |
| **Mistral-7B-Instruct-v0.3**, *KV Budget = Full* | | | | | | | | | | | | | | | |
| Full KV | 29.20 | 38.27 | 50.09 | 51.20 | 36.49 | 26.67 | 34.21 | 25.79 | 26.43 | 76.00 | 88.59 | 47.46 | 59.37 | 60.52 | 46.45 |
| **Mistral-7B-Instruct-v0.3**, *KV Budget = 128L* | | | | | | | | | | | | | | | |
| SnapKV | 23.77 | 28.60 | 45.64 | 44.60 | 29.85 | 21.92 | 22.66 | 22.56 | 21.32 | 47.50 | 87.94 | 43.32 | 54.67 | 51.75 | 39.00 |
| AdaKV | 24.54 | 29.42 | 45.65 | 42.40 | 28.66 | 22.18 | 22.89 | 23.36 | 21.53 | 65.50 | 88.89 | 44.55 | 53.89 | 53.23 | 40.18 |
| FastKV | 24.97 | 26.68 | 45.46 | 43.75 | 30.01 | 21.06 | 22.43 | 22.33 | 20.37 | 63.50 | 89.24 | 42.50 | 54.37 | 51.80 | 39.89 |
| CAKE | 22.31 | 29.15 | 43.51 | 44.51 | 30.36 | 22.85 | 21.56 | 20.47 | 18.96 | 47.00 | 88.60 | 39.36 | 44.96 | 46.19 | 37.13 |
| TAKE | 23.25 | 28.56 | 43.64 | 48.80 | 34.32 | 20.92 | 22.06 | 21.86 | 25.63 | 56.50 | 89.16 | 43.19 | 58.40 | 51.97 | 40.43 |
| **Mistral-7B-Instruct-v0.3**, *KV Budget = 256L* | | | | | | | | | | | | | | | |
| SnapKV | 24.72 | 31.23 | 47.01 | 46.72 | 31.09 | 22.74 | 25.08 | 23.60 | 22.82 | 62.00 | 88.99 | 45.11 | 56.82 | 56.39 | 41.74 |
| AdaKV | 25.88 | 31.39 | 46.56 | 46.47 | 31.67 | 22.21 | 24.18 | 23.93 | 22.88 | 72.00 | 89.16 | 45.17 | 56.71 | 56.97 | 42.51 |
| FastKV | 24.75 | 29.61 | 49.27 | 48.33 | 31.91 | 24.13 | 24.45 | 22.80 | 22.60 | 70.00 | 89.17 | 43.65 | 56.69 | 55.05 | 42.31 |
| CAKE | 24.01 | 30.78 | 45.88 | 45.06 | 31.45 | 23.43 | 22.81 | 20.87 | 20.92 | 58.50 | 88.29 | 40.16 | 47.76 | 48.43 | 39.17 |
| TAKE | 24.82 | 32.47 | 48.26 | 50.05 | 34.64 | 22.33 | 24.18 | 22.81 | 25.85 | 68.50 | 89.34 | 44.90 | 59.37 | 55.64 | 43.02 |
| **Mistral-7B-Instruct-v0.3**, *KV Budget = 512L* | | | | | | | | | | | | | | | |
| SnapKV | 26.03 | 34.82 | 49.74 | 49.25 | 34.11 | 23.88 | 27.23 | 24.33 | 24.59 | 74.00 | 89.82 | 45.94 | 58.13 | 58.49 | 44.31 |
| AdaKV | 27.14 | 35.10 | 48.65 | 48.98 | 34.30 | 23.80 | 25.76 | 24.38 | 24.27 | 75.00 | 89.14 | 45.87 | 58.18 | 59.49 | 44.29 |
| FastKV | 25.94 | 33.20 | 49.03 | 48.37 | 34.42 | 25.43 | 26.63 | 23.88 | 24.10 | 71.50 | 89.27 | 46.49 | 58.72 | 58.63 | 43.97 |
| CAKE | 25.82 | 34.24 | 46.93 | 45.34 | 31.71 | 23.38 | 25.81 | 21.47 | 22.41 | 68.00 | 88.46 | 41.27 | 49.04 | 49.67 | 40.96 |
| TAKE | 26.65 | 35.61 | 49.49 | 50.11 | 37.42 | 22.99 | 26.18 | 24.09 | 25.98 | 73.50 | 89.54 | 45.12 | 59.86 | 58.18 | 44.62 |
| **Mistral-7B-Instruct-v0.3**, *KV Budget = 1024L* | | | | | | | | | | | | | | | |
| SnapKV | 28.31 | 36.57 | 48.21 | 50.57 | 35.02 | 24.71 | 29.22 | 24.86 | 26.29 | 75.00 | 88.99 | 46.08 | 58.76 | 60.24 | 45.20 |
| AdaKV | 26.91 | 36.41 | 48.78 | 50.02 | 35.76 | 24.85 | 27.43 | 24.86 | 25.55 | 75.50 | 89.61 | 46.92 | 58.93 | 60.07 | 45.10 |
| FastKV | 26.43 | 36.69 | 49.44 | 49.27 | 35.04 | 25.69 | 28.63 | 24.77 | 25.83 | 73.50 | 88.64 | 47.34 | 59.13 | 59.63 | 45.00 |
| CAKE | 26.09 | 36.34 | 48.11 | 45.97 | 32.39 | 23.49 | 27.56 | 21.45 | 24.03 | 72.50 | 88.61 | 42.71 | 51.06 | 51.25 | 42.25 |
| TAKE | 26.55 | 37.1 | 50.26 | 51.46 | 37.93 | 23.94 | 28.56 | 24.64 | 26.29 | 75.00 | 89.71 | 45.39 | 59.77 | 58.4 | 45.36 |

## D  LLM USAGE STATEMENT

During the preparation of this manuscript, we utilized Large Language Models (e.g., GPT-5) as a general-purpose writing assistant. The use of LLMs was primarily for improving grammar, spelling, and clarity. We reviewed and edited all suggested changes and take full responsibility for the final content of this paper.

