# OpenReview forum: "TAKE: Task-Aware Chunked KV Cache Eviction for Efficient Long-Context LLM Prefill"
_ICLR.cc/2026/Conference — Submitted to ICLR 2026_

### Official Review · Reviewer_UKFZ · 2025-10-21

**Soundness:** 2
**Presentation:** 3
**Contribution:** 2
**Rating:** 4
**Confidence:** 4

**Summary:**

This paper introduces TAKE, a training-free framework designed to reduce memory usage and latency during the prefill stage of long-context LLM inference. The core idea is to process the input sequence in chunks, interleaving attention computation with a novel KV cache eviction strategy. TAKE utilizes two main components: an "accumulated task-aware probe" to identify and retain task-relevant tokens across chunks, and a "delayed eviction" strategy to protect shallow layers from premature information loss. Experiments show that TAKE significantly lowers peak memory usage and time-to-first-token (TTFT) while maintaining strong performance on long-context benchmarks.

**Strengths:**

- The paper is well-structured and easy to follow. The figures are effective at illustrating the core concepts of chunked processing and the overall pipeline, which helps in understanding the proposed method.
- The ablation studies are relatively comprehensive. The authors have made an effort to validate the effectiveness of the two main components of their method (the task-aware probe and delayed eviction), which provides valuable insights into why TAKE works.

**Weaknesses:**

- The proposed method appears to be a chunked version of an importance-based eviction method like SnapKV. For a more direct and convincing comparison, it would be beneficial to adapt SnapKV to a similar chunked processing framework. This would help isolate and verify the true effectiveness of the proposed probe mechanism over existing attention-based scoring functions in a chunked setting.

- The method's reliance on using only the probe to score keys seems to create a high dependency on the accuracy of probe selection.

- The paper lacks references to some relevant prior work on chunk-based processing for KV cache management. SARATHI[1] proposed the concept of chunked-prefill. OmniKV[2] also employs a chunking strategy to accelerate prefill and minimize the KV cache for H2O.

[1] SARATHI: Efficient LLM Inference by Piggybacking Decodes with Chunked Prefills

[2] OmniKV: Dynamic context selection for efficient long-context LLMs

**Questions:**

1.  In the ablation study (Table 2), for the "w/o probe" configuration, what scoring method was used to select which keys to evict? The paper states that this configuration performs poorly, but it is unclear what alternative scoring function was used in the absence of the probe.

2.  The current probe selection strategy uses the last few tokens of the input, which works well for benchmarks like Needle-in-a-Haystack. However, for other tasks like style transfer, where instructions might appear at the beginning (e.g., `{Reference Style}{Instruction}{Article to Convert}`), would it be more robust to scan the context for explicit task instructions instead of relying on token position?

3.  For a complex multi-hop question where the probe itself contains multiple clauses and dependencies, how confident are the authors that this method can still effectively identify and retain all the necessary pieces of evidence scattered across different chunks?

4.  Could the analysis in Figure 7b be expanded along the "Warm-up Layers" dimension? It would be interesting to see a clearer trend of how performance changes as the number of warm-up layers increases or decreases for the LongBench tasks.

---

> ### Author Response · Authors · 2025-11-25
>
> We thank all reviews for their insightful comments, and constructive questions, which are extremely helpful for improving our work.
> Our responses to your questions are as follows:
>
> # W1: Adapt SnapKV to proposed method
> A: Our proposed TAKE adopts the pooling operation in SnapKV to smooth token importance scores. The performance of SnapKV without probe tokens can be approximated by the Chunk-only variant in our ablation study, as shown in Table 2. It is equivalent to a Chunked SnapKV setup where the importance of each token is computed using the query formed by the last tokens of the current context.
>
> # W2: Dependency on the accuracy of probe selection
> A: In our benchmark setup, each prompt consists of two parts: the context and the task instruction. The instruction segment is exactly the text from which we construct the probe tokens. In practice, we insert special markers into the prompt template to locate the task instruction, so that we can accurately extract the instruction span and use it to build the probe tokens.
>
> # W3: The citation of SARATHI and OmniKV
> A: We cite SARATHI in the PRELIMINARY section (specifically, line 125), where we introduce the background of chunked prefill and provide a corresponding proof in the appendix. OmniKV, in essence, still belongs to the family of global KV pruning methods. In OmniKV, a 'chunk' is the unit of data transfer between CPU and GPU, and in TAKE, the unit of forward propagation over the input text, which are different definitions. We will add OmniKV to the related work section, but it is not necessary to include it as a baseline for further comparison. In our comparisons, we have already included strong and representative global KV pruning baselines.
>
>
> # Q1: Scoring method in "w/o probe" configuration
> A: In the w/o Probe setting, we use the queries of the last tokens in the current chunk (with the same length as the probe tokens) to select important tokens. Since the current chunk lacks task-related information, it is difficult to identify task-critical tokens under chunked prefill, presenting weaker performance without probe tokens.
>
> # Q2: Probe selection strategy
> A: Thanks for your suggestion. The learnable components, as well as scanning the context, can be considered to generate probe tokens, but they bring extra inference overhead. We will explore a more efficient way to select probe token. TAKE relies on the prompt template to locate and extract task-related tokens. This is the most cost-effective way to obtain task-relevant tokens.
>
>
> # Q3: The ability of method for multi-hop question
> A: Thanks! The LongBench benchmark already includes multi-hop QA tasks such as HotpotQA. It is observed that with a KV cache size of 512L, the performance gap compared to the full model is very small. Since TAKE performs chunk-wise iterative inference, it naturally supports multi-hop reasoning. Probe tokens are continuously fused across chunks, and the evaluation query is updated accordingly. Therefore, the model can effectively retrieve key information in multi-hop QA scenarios.
>
>
> # Q4: Analysis in Figure 7b along the "Warm-up Layers" dimension
> A: In Figure 7b, the trend with respect to `warmup_layers` does not appear to be “the more, the better”, since different task types have different optimal choices, and LongBench is a composite benchmark. For example, in retrieval-style tasks, attention sparsity emerges in relatively shallow layers. Reducing `warmup_layers` in such cases shrinks the KV cache in early layers and reduces redundancy during forward propagation. Empirically, we find that setting `warmup_layers` to about half of the model depth performs well across most tasks. While this choice is not always optimal for every single task, it is the most stable setting overall.

---

> ### Author Response · Authors · 2025-12-03
>
> Dear Reviewer,
>
> I hope this message finds you well. As the discussion period is nearing its end with less than three days remaining, I wanted to ensure we have addressed all your concerns satisfactorily. If there are any additional points or feedback you'd like us to consider, please let us know. Your insights are invaluable to us, and we’re eager to address any remaining issues to improve our work.
>
> Thank you for your time and effort in reviewing our paper.

---

### Official Review · Reviewer_RBkD · 2025-10-23

**Soundness:** 3
**Presentation:** 3
**Contribution:** 2
**Rating:** 4
**Confidence:** 5

**Summary:**

This paper proposes TAKE, a KV cache–centric optimization that performs KV cache eviction together with chunked prefill to reduce peak GPU memory usage during LLM inference. TAKE primarily introduces two mechanisms: accumulated task-aware probing and delayed eviction. The accumulated task-aware probe estimates the relevance between each KV pair and the task query by applying an exponential moving average (EMA) over the query (Q) vectors during chunked prefill. The probe queries are drawn from the trailing tokens of the entire input sequence, as these tokens typically contain task-related information. They are smoothed during chunked prefill because query tokens are often strongly biased toward the current chunk. The delayed eviction mechanism maintains a relaxed eviction budget for shallow layers, and reduces it to the target budget when processing the final chunk. The positions preserved in shallow layers are determined by the first layer that enables eviction. TAKE is evaluated on multiple benchmarks, including NIAH and LongBench, and compared with various baselines. Empirical results show that TAKE achieves nearly lossless performance while providing the fastest TTFT among all baselines.

**Strengths:**

1. This paper is well written and easy to follow. The description of the method and the two major contributions are clear. The benchmark results demonstrate the promising performance and efficiency of TAKE.

2. The design of TAKE is reasonable, and the ablation studies explain the effectiveness of the proposed mechanisms. Conducting KV cache eviction along with chunked prefill can significantly reduce peak memory usage without notably harming performance.

3. TAKE provides a promising method to host a long-context LLM in GPU-memory-constrained scenarios, e.g., consumer-grade devices. The contribution of this paper is sufficient.

**Weaknesses:**

1. Although combining chunked prefill with KV cache eviction is an efficient design, the novelty remains limited. There are many related papers that work in this area. For example, InfiniPot [1] and Locret [2] also integrate eviction with chunked prefill, and they both achieve good performance on downstream long-context tasks. These baselines should be compared with TAKE as the main baselines, as they address the same problem.

2. NIAH is a relatively simple long-context retrieval task, and the input sequence length of LongBench is limited (most entries are less than 40K). Therefore, a more challenging benchmark is expected to be tested. RULER [3] is an appropriate benchmark for evaluating TAKE. Since query-aware KV selection is introduced in TAKE, it should (or is expected to) achieve nearly lossless performance on RULER. Please include this experiment in your paper, and discuss the potential reasons if there is a large performance degradation.

3. Multi-turn conversation is also an important task for long-context processing. Since query-irrelevant tokens are evicted in the design of TAKE, is it able to process multi-turn conversations without re-prefilling the previous chat turns? A brief discussion should be included in the paper, and if possible, discuss the potential limitations of scenarios that TAKE cannot handle. Empirical results are also welcome to clarify this issue. Since multi-turn conversation is mentioned in the introduction section (line 37), such discussion is especially necessary.

4. One possible way to reduce peak memory is to run SnapKV and conduct layer-wise eviction. In other words, the model first prefills a certain layer, then performs eviction using SnapKV before executing the computation of the next layer. Such baseline methods should also be discussed. One advantage of TAKE is that it can process extremely long sequences, where even a single-layer KV cache can exceed GPU memory constraints—for example, conducting retrieval tasks at a 10M-token input scale. Discussing such scenarios can further demonstrate the superiority of TAKE compared with vanilla methods.


---

[1] InfiniPot: Infinite Context Processing on Memory-Constrained LLMs

[2] Locret: Enhancing Eviction in Long-Context LLM Inference with Trained Retaining Heads

[3] RULER: What's the Real Context Size of Your Long-Context Language Models?

**Questions:**

See above.

---

> ### Author Response · Authors · 2025-11-28
>
> We thank all reviews for their insightful comments, and constructive questions, which are extremely helpful for improving our work. Our responses to your questions are as follows:
>
> # Q1: Comparison with InfiniPot and Locret
>
> A: Thanks! InfiniPot and Locret are indeed both methods that combine chunked prefill and we will add them in related works. InfiniPot uses a combination of attention scores and cross-entropy to determine token importance.
> - However, based on InfiniPot's LongBench evaluation results, for the same Llama3.1-8B-Instruct model, InfiniPot with a 4k KV cache size performs significantly worse than TAKE with only a 0.5k KV cache size. We attribute this to TAKE's novel delayed eviction strategy, which results in less information loss. As we demonstrate in Figure 4, shallow layers exhibit insufficient attention sparsity. InfiniPot adopts a layer-consistent approach that ignores the significant information loss caused by pruning when attention distributions are smooth. Due to the forward dependency in chunked prefill, this loss accumulates progressively across chunk iterations. This is why TAKE achieves performance closer to the original model and even surpasses the native model on some long document retrieval and QA tasks.
>
>     |Method|Training-free|KV Budget|Longbench Score|
>     |--|--|--|--|
>     |TAKE|Yes|512|47.45|
>     |InfiniPot|Yes|4096|42.75|
>
> - Locret adds lightweight learnable retaining heads to each attention layer that predict a causal importance score for every KV cache unit, trained to mimic the effect of full attention and thus estimate how much each token will matter for future generation. However, Locret requires additional training and MLP layers to determine token importance, making it neither plug-and-play nor efficient due to extra computational overhead. Our focus is on training-free methods with high efficiency and low cost, so direct comparison would not be fair.
>
>
>
>
> # Q2: Evaluation on RULER benchmark
> A: Thanks! We have supplemented our evaluation with RULER benchmark results as shown below. We observe a performance drop on RULER compared to LongBench, which we attribute to the specific characteristics of the needle-in-haystack tasks in RULER. In these tasks, the "needle" consists of semantically meaningless strings (e.g., random character sequences), whereas TAKE relies on semantic information to identify and retain task-relevant tokens. When the critical information lacks semantic coherence, our probe token mechanism cannot effectively distinguish it from the surrounding context, leading to potential eviction of important tokens. This reveals a limitation of TAKE in scenarios where key information is semantically arbitrary. We thank the reviewer for helping us identify this constraint, and we plan to explore more robust token selection strategies that can handle both semantic and non-semantic critical information in future work.
>
> |Method|4k|8k|16k|32k|
> |--|--|--|--|--|
> |TAKE-2048|94.35|47.45|30.18|27.45|

---

> ### Author Response · Authors · 2025-11-28
>
> Responses to remaining questions:
>
> # Q3: Extension to multi-turn dialogue scenarios and our limitation
>
> A: Thanks! TAKE can be extended to multi-turn dialogue scenarios with minor modifications. Considering that each user request may focus on different parts of the long context, we need to offload the complete KV cache to CPU as a backup. When a user request arrives, we recall the KV cache chunk by chunk and apply the same token selection method described in the paper to retain task-relevant tokens in GPU memory. This residency is specific to the current request; when the next request arrives, KV selection needs to be performed again based on the new query.
>
> However, we acknowledge a potential limitation: repeatedly performing KV selection for each turn may introduce additional latency, especially when user queries shift focus across different context regions.
> A more sophisticated caching strategy could maintain a compact set of globally important tokens across turns while dynamically adjusting based on new queries. We believe this represents an interesting direction for future work on KV eviction-based LLM acceleration in multi-turn dialogue settings. Due to the complexity of designing and evaluating such multi-turn scenarios (requiring diverse conversation patterns and context-switching behaviors), we leave a comprehensive experimental study of this extension to future research.
>
> # Q4: KV cache pruning after attention computation and ability to deal with long-context input scale
>
> A: Thanks! Among our baseline methods, FastKV adopts a similar idea to yours, but FastKV prunes hidden states at the middle layers of the model to reduce the input to subsequent transformer blocks, which naturally reduces the KV cache size from that layer onward. However, FastKV still processes hidden states of the full sequence in shallow layers, introducing higher peak memory usage. As shown in the table below, TAKE maintains a stable and minimal memory footprint across different sequence lengths, while both FullKV and FastKV exhibit significant memory growth as sequence length increases:
> | Sequence length | 16K | 32K | 64K | 128K |256K|
> | :-- | :-- | :-- | :-- | :-- | :-- |
> | FullKV | 20.20 | 24.30 | 32.50 | 48.91 |OOM|
> | FastKV| 18.08	|20.03	| 23.95 |	31.77	| 47.41 |
> | TAKE (Ours) | 18.98 | 18.98 | 18.99 | 18.99 |19.00|
>
> We also find that the KV size in shallow layers is critical for model performance but still apply KV eviction on shallow layers to control peak VRAM usage. If resources permit, retaining the complete KV cache in shallow layers can better preserve model performance. However, for extremely long sequences, even a single layer's KV cache size may exceed GPU memory constraints. Therefore, TAKE employs delayed eviction to balance memory usage and model capability. For shallow layers that require retaining more KV cache, we preserve a relatively larger KV size during the chunk iteration process and only prune to the target size at the final step. The parameter `warm-up budget` reflects the impact of shallow-layer KV cache size on model performance during the prefill process. The detailed results of LongBench under different warmup-budget are as follows:
>
> |warm-up budget|LongBench Score|
> |--|--|
> |1024|46.07|
> |2048|47.00|
> |4096|47.45|
> |8192|47.52|
>
> Finally, we thank you for pointing out both the advantages and limitations of TAKE and providing valuable suggestions, which are very helpful for our research.

---

> ### Author Response · Authors · 2025-12-03
>
> Dear Reviewer,
>
> I hope this message finds you well. As the discussion period is nearing its end with less than three days remaining, I wanted to ensure we have addressed all your concerns satisfactorily. If there are any additional points or feedback you'd like us to consider, please let us know. Your insights are invaluable to us, and we’re eager to address any remaining issues to improve our work.
>
> Thank you for your time and effort in reviewing our paper.

---

### Official Review · Reviewer_SHwH · 2025-10-27

**Soundness:** 2
**Presentation:** 2
**Contribution:** 3
**Rating:** 4
**Confidence:** 4

**Summary:**

This paper proposes TAKE, a dynamic KV-cache Eviction during the chunked-prefilling process to reduce memory usage and TTFT (time-to-first-token) and enable 128K inference on 24GB GPUs. The idea is novel in chunk-wise KV eviction, while a similar idea has been deployed on decoding, including SnapKV and R-KV. The Training-Free design and will-design experiment on Llama3 and Mistral, while the missing reasoning model experiments of DeepSeek or Qwen are also a big question as to whether TAKE is also effective on the reasoning model.

**Strengths:**

Unlike global pruning methods that prune after full prefill, such as SnapKV, TAKE prunes during prefill to avoid high memory peaks. it would help for long-context prefill tasks for both throughput and memory usage.

It uses smart semantic preservation to retain necessary tokens while maintaining accuracy. The accuracy looks good, and it is important for real-world application that requires high accuracy.

We believe training-free is essential for effective KV Cache Eviction, which this paper achieves.

**Weaknesses:**

Although it shows novelty compared to chunked prefill, we have observed a similar idea on decoding and gradually pruning the KV cache during decoding. If we set a chunk size during decoding and raise a token pruning every chunk size, the idea can also be deployed on a reasoning model and decoding process, which would significantly improve the impact of the pruning as reasoning models like DeepSeek and Qwen are becoming more and more important now.

An essential concept of chunk prefill is the selection of the chunk size; different chunk sizes affect the prefill process's performance. And more importantly, it should also affect the performance of TAKE Task-Aware Chunked KV Cache Eviction. However, I didn't see any discussion of the chunk size in the paper (even worse, I can't see the chunk size number in the main paragraph; there's only one discussion in Appendix A about setting the chunk size Z = 4096). This is an important issue that changed my rating from 6 to 4. I would like to see the ABLATION STUDY on how the chunk size affects memory usage, prefill time, and the accuracy of TAKE.

LongBench can be updated to LongBench v2 for state-of-the-art experiments.

Typo:

line 53: remaining -> remain

line 367:  time-to-fist-token -> time-to-first-token

**Questions:**

what is the maximum context length TAKE can support on RTX-4090, 128K or higher?

---

> ### Author Response · Authors · 2025-11-27
>
> We thank all reviews for their insightful comments, and constructive questions, which are extremely helpful for improving our work. Our responses to your questions are as follows:
>
> # W1: Applying the method in the decoding stage
>
> A: Thanks! While TAKE can indeed be applied in the decoding stage by setting a threshold to maintain a fixed KV cache size, in long-context reasoning tasks the number of output tokens is typically much smaller than the number of input tokens. Our work therefore focuses on introducing pruning in the prefill stage, which has the greatest impact on long-context inference, to substantially reduce peak memory usage. This allows the prefill phase to also exploit attention sparsity to improve inference performance. Moreover, in TAKE the decoding stage also benefits from the prefill-stage optimization, since by that point the KV cache has already been reduced to a much smaller size.
>
> # W2: Impact of different chunk sizes
>
> A: Thanks! We present these results in the table below, which indicate that chunk size has only a minor effect on model performance and primarily influences resource consumption. Larger chunks require more GPU memory, while smaller chunks are more suitable for resource-constrained devices to avoid OOM errors. We do not observe a clear difference in TTFT: smaller chunks reduce the computation per step but increase the number of CUDA kernel launches, so the overall latency does not change significantly.
>
>
> | Chunk Size | Peak VRAM | Inference Latency | Average Score |
> | :-- | :-- | :-- | :-- |
> | 2048 | 17.55GB | 27.8s | 47.11 |
> | 4096 | 18.22GB | 28.1s | 47.39 |
> | 8192 | 20.32GB | 28.6s | 47.23 |
>
> # W3: Evaluation on LongBench v2
>
> A: Thanks! We will supplement our evaluation with LongBench v2 results. The results are as follows:
>
>
> | Length | Full | TAKE-2048 |
> | :-- | :-- | :-- |
> | Short | 31.1 | 27.8 |
> | Medium | 27.8 | 21.7 |
> | Long | 23.1 | 18.5 |
>
> We find that LongBench v2 is a much harder benchmark than LongBench v1,so we set a larger kv budget:2048$L$, as other researchers do, to achieve similar performance with full KV.
>
> # W4: Grammatical errors
>
> A: Thanks for pointing out the grammatical errors. We will make the necessary corrections.
>
> # Q1: Maximum length supported by RTX-4090
>
> A: Thanks! TAKE can in principle support arbitrary context length as long as the model does. If the model supports a 1M-token context, then for an input of length 1M the memory usage is essentially the same as for 128k or 64k. By chunking the input, we keep the KV cache footprint small throughout the entire prefill stage, as explained in lines 422–427. As shown in the table, the memory usage of our method remains stable at around 19GB, increasing with the sequence length from 16K to 256K. FullKV shows serious growth, while other baseline methods based on global KV eviction also show steady growth(the peak VRAM of other methods is nearly identical, so we use their average value for comparison, with their difference not exceeding 0.2GB). It demonstrates that our method can support contexts of arbitrary length once the device, such as RTX-4090, can accommodate the model's basic parameters and the foundation model supports valid outputs of arbitrary sequence length.
>
> | Sequence length | 16K | 32K | 64K | 128K |256K|
> | :-- | :-- | :-- | :-- | :-- | :-- |
> | FullKV | 20.20 | 24.30 | 32.50 | 48.91 |OOM|
> | Baseline methods (Avg.)| 18.24 | 20.19 |24.09 | 31.90 |47.52|
> | TAKE (Ours) | 18.98 | 18.98 | 18.99 | 18.99 |19.00|
>
> TAKE's memory usage is related to the chunk size, as shown in the table in W2 (Impact of different chunk sizes). Memory usage increases with chunk size, but not exponentially.
> This is the key advantage of TAKE: chunk-wise eviction significantly reduces the peak memory during prefill, while probe tokens and delayed eviction mitigate the performance degradation that chunking would otherwise introduce.

---

> > ### Comment · Reviewer_SHwH · 2025-11-27
> >
> > thank for your reply. what is other methods mean?

---

> ### Author Response · Authors · 2025-11-27
>
> Thanks for your reply.
>
> Other methods refer to baseline methodes that all adopt global KV eviction strategies,including SnapKV, AdaKV and FastKV.
> Detailed results  of memeory usag are shown in the table below.
> | Sequence length | 16K | 32K | 64K | 128K |256K|
> | :-- | :-- | :-- | :-- | :-- | :-- |
> | FullKV | 20.20 | 24.30 | 32.50 | 48.91 |OOM|
> | SnapKV| 18.34 | 20.27 |24.18 | 32.00 |47.64|
> | AdaKV| 18.30 | 20.26 |24.15 | 31.94 |47.51|
> | FastKV| 18.08 | 20.03 |23.95 | 31.77 |47.41|
> | TAKE (Ours) | 18.98 | 18.98 | 18.99 | 18.99 |19.00|

---

> > ### Comment · Reviewer_SHwH · 2025-11-27
> >
> > Thank authors for solving my concern. I have raised my score to 6.

---

### Meta-Review · Area_Chair_yXqj · 2026-01-05

**Summary:**

The paper proposes TAKE - a training free framework for task-aware KV cache pruning during chunked prefill. The proposed technique significantly reduces GPU usage and time-to-first-token for long-context LLM inference. All reviewers agreed that the problem is very practical, and that the solution is motivated well enough. Initial concerns focused on novelty, lack of comparison to other similar methods, missing ablations. The rebuttal helped one of the reviewers to increase the score from 4 to 6. However, given the other scores to be 4, and the overall close to borderline scores, I recommend this paper for rejection.

**Reviewer Concerns:**

The rebuttal added multiple new benchmarks and ablations, such as LongBenchv2 and RULER. Multiple missing baselines were added such as SnapKV. One of the reviewers participated in the discussion and was considering changing the score from 4 to 6 due to new ablations, longbench v2 results. Other 2 reviewers didnt participate in the discussion, and it is hard to tell if concerns were addressed or not as they are primarily centered around novelty.

**Reviewer Scores:**

From initial scores of all 4, one of the reviewers was considering changing to 6, the other 2 didnt participate in the discussion, and might not be satisfied with the novelty aspect.

---

### Decision · Program_Chairs · 2026-01-26

Reject